# Oxidative Stress and Neuroinflammation in Parkinson’s Disease: The Role of Dopamine Oxidation Products

**DOI:** 10.3390/antiox12040955

**Published:** 2023-04-18

**Authors:** Sasanka Chakrabarti, Marco Bisaglia

**Affiliations:** 1Department of Biochemistry and Central Research Laboratory, Maharishi Markandeshwar Institute of Medical Sciences and Research, Maharishi Markandeshwar University (Deemed to be), Mullana, Ambala 133207, India; sasankac@mmumullana.org; 2Department of Biology, University of Padova, Via Ugo Bassi 58/B, 35131 Padova, Italy; 3Study Center for Neurodegeneration (CESNE), 35121 Padova, Italy

**Keywords:** dopamine, DOPAL, neuromelanin, oxidative stress, Parkinson’s disease, synuclein

## Abstract

Parkinson’s disease (PD) is a chronic neurodegenerative condition affecting more than 1% of people over 65 years old. It is characterized by the preferential degeneration of nigrostriatal dopaminergic neurons, which is responsible for the motor symptoms of PD patients. The pathogenesis of this multifactorial disorder is still elusive, hampering the discovery of therapeutic strategies able to suppress the disease’s progression. While redox alterations, mitochondrial dysfunctions, and neuroinflammation are clearly involved in PD pathology, how these processes lead to the preferential degeneration of dopaminergic neurons is still an unanswered question. In this context, the presence of dopamine itself within this neuronal population could represent a crucial determinant. In the present review, an attempt is made to link the aforementioned pathways to the oxidation chemistry of dopamine, leading to the formation of free radical species, reactive quinones and toxic metabolites, and sustaining a pathological vicious cycle.

## 1. Introduction

Parkinson’s disease (PD) is a chronic progressive neurodegenerative disorder affecting more than 1% of people over the age of 65 years. Motor impairments, and in particular, resting tremors, postural instability, muscular rigidity, and bradykinesia, represent the main clinical symptoms of the disorder, although non-motor complications, such as depression, sleep disorders, olfactory impairment, and constipation, can be present as well. Locomotor dysfunctions result from the preferential degeneration of dopaminergic neurons in the substantia nigra pars compacta, which represents one of the pathological hallmarks of the disease. This leads to a depletion of dopamine (DA) at the striatal level, where the neurotransmitter is essential to control movement [1]. Another pathological feature of the disease is the presence of cytosolic inclusions referred to as Lewy bodies (LB), which are mainly composed of α-synuclein (αSyn) [2]. Importantly, further highlighting the role of the protein in PD onset, mutations in the gene *SNCA* coding for αSyn, as well as gene duplications and triplications, have been found in genetic forms of the disorder [3,4].

PD is still an incurable disorder, and the current therapeutic treatments can only provide symptomatic relief, while they are unable to stop or hamper the progression of the disease. The gold standard treatment for PD is L-DOPA, a DA precursor, which is able to cross the blood–brain barrier. L-DOPA is often administered in combination with peripheral L-DOPA decarboxylase inhibitors, which prevent the conversion of L-DOPA into DA before its brain accumulation. L-DOPA therapy is initially effective in ameliorating symptoms in most patients, but its long-term administration is often accompanied by several side effects, such as dyskinesias and wearing off responses, some of which could rely on DA intrinsic toxicity, as discussed below.

PD is considered to be a multifactorial disorder, in which genetic susceptibilities and environmental factors contribute to the onset of the disease [5,6]. While the etiology of PD is still partially elusive, proteostasis alterations, mitochondrial dysfunctions, oxidative stress, and neuroinflammation are generally recognized as key factors involved in the pathology [7,8,9]. The coexistence of these processes, with nigral cell loss in PD brains and experimental PD models, makes it difficult to establish whether they (or some of them) are the cause of nigral degeneration, or rather a consequence. What is clear, on the contrary, is that all these processes appear to be strictly correlated and influence each other. Since the discovery of high levels of oxidized DNA, proteins, and lipids and lower levels of glutathione (GSH) in post-mortem PD brains, special interest has been attributed to the role of oxidative damage in PD [7]. In this frame, the iron-catalyzed Fenton’s reaction and the breakdown of lipid-derived alkoxy and peroxy radicals contribute to cellular oxidative stress, and an increased accumulation of iron in the substantia nigra of PD brains has been reported from post-mortem studies and by ante-mortem imaging techniques [10,11]. As mitochondria are considered to be a primary source of reactive oxygen species (ROS), mitochondrial dysfunctions are believed to abundantly participate in driving the redox alterations observed in the disease [12]. Coherently, numerous PD-related genes and neurotoxins have been shown to affect mitochondrial functionality [13]. Moreover, several mitochondrial constituents and metabolites can function as damage-associated molecular patterns (DAMPs) and stimulate inflammatory responses when they are released into the cytosol or extracellular environment [14,15]. At the same time, through the chronic activation of microglia, neuroinflammation can be responsible for a conspicuous production of ROS, which may contribute to amplifying oxidative conditions [14,16]. In fact, microglial activation may lead to increased ROS production through the activation of NADPH oxidase (NOX), which is an important source of cellular ROS, and the upregulation of NOX was reported both in post-mortem brains of PD patients and toxin-induced animal models of PD [17,18]. Interestingly, gut dysbiosis has been recently suggested to participate in the onset and/or progression of PD, and alterations in the gut microbiota population have been proposed to lead to gut permeability and enhance the systemic inflammatory response. The long-term activation of peripheral immune cells may, in turn, promote the breakage of the blood–brain barrier, leading to chronic neuroinflammation [19,20].

While redox alterations, mitochondrial dysfunctions, and neuroinflammation are clearly involved in PD pathology, how these processes lead to the preferential degeneration of dopaminergic neurons is still an unanswered question. In this context, as represented in Figure 1, the presence of DA itself within this neuronal population could represent a crucial determinant.

In this review, we will take into consideration the cellular toxicity associated with DA dyshomeostasis in an attempt to rationalize the preferential degeneration of specific neuronal populations, which characterizes PD.

## 2. DA Metabolism and Oxidative Chemistry

As a neurotransmitter, DA plays a central role in several mental and physical functions, including motivation, reward, sleep, mood, attention, and voluntary movement [21]. In light of the numerous functions exerted by DA, its levels must be strictly regulated, and an elevated concentration of this neurotransmitter, as well as its deficiency, may be associated with pathological conditions. As represented in Figure 2, DA is synthesized in the cytosol through the action of tyrosine hydroxylase (TH), which converts the amino acid tyrosine into 3,4-dihydroxyphenylalanine (L-DOPA), which is then transformed into DA by the action of aromatic amino acid decarboxylase (AADC). Alternatively, after its release into the synaptic cleft, DA can be reabsorbed from the extracellular space into the cytosol by the action of the DA transporter (DAT), a membrane-spanning protein that couples the electrochemical favorable movement of Na^+^ ions to co-transport DA across the plasma membrane. Once it is in the cytosol, DA is transferred inside synaptic vesicles where it is stabilized by the luminal low pH by the vesicular monoamine transporter 2 (VMAT2) in another ATP-dependent process [21]. In physiological conditions, cytosolic DA can be metabolized via monoamine oxidase at the level of the outer mitochondrial membrane to form the 3,4-dihydroxyphenilacetldehyde (DOPAL), a highly toxic molecule that, in turn, is converted to the non-toxic metabolite 3,4-dihydroxyphenylacetic acid (DOPAC) by the enzyme aldehyde dehydrogenase. DA metabolism can also happen at the astrocytic level where the enzymes involved in the DA-degradation pathways are equally expressed [21,22]. However, the very low expression levels of the DAT in the astrocytes of the nigrostriatal pathway make the contribution of astrocytes to this process very limited in both the striatum and substantia nigra [22].

At the neutral pH that characterizes the cytosolic environment, DA is relatively unstable, and it can auto-oxidize. As a consequence, the accumulation of DA in the cytosol, due to an impairment in either its synthesis, accumulation inside synaptic vesicles, catabolism, or reuptake from the synaptic cleft can promote its spontaneous oxidation, leading to the generation of both ROS and dopamine quinones (DAQs), which are highly reactive electrophilic molecules. In fact, the autoxidation of DA has been demonstrated to generate both superoxide anion and hydrogen peroxide [23], while the in vivo relevance of the last pathway is corroborated by the presence inside dopaminergic neurons of neuromelanin (NM), a dark pigment mainly composed of polymerized DA, proteins, and lipids [24,25,26].

## 3. DA-Related Toxicity

Both ROS and DAQs can induce deleterious cytotoxic effects, promoting neuronal degeneration. It is now clear that while low levels of ROS might exert physiological functions, excessive levels of ROS can damage cellular components such as proteins, lipids, and DNA [7]. At the same time, the electron-deficient DAQs can react with and covalently modify cellular nucleophiles, and the cysteine residues, which are often found in the active sites of proteins, are considered to be their principal targets [7]. DA oxidation-derived toxicity in vivo was first demonstrated by the injection of high concentrations of DA into rats’ striatum [27]. More recently, the same group used a more sophisticated model based on the viral-mediated interference of VMAT2 expression in adult rats to induce dopaminergic dysregulation. The loss of VMAT2 resulted in the accumulation of cytosolic DA with a corresponding increase in DA turnover and in DAQs formation, as measured by protein cysteinyl-DA adducts, coupled with deficits in DA-mediated behaviors, and the degeneration of nigrostriatal dopaminergic neurons [28]. A similar conceptual approach to induce DA accumulation was used by another group that over-expressed functional TH in the endogenous, restricted catecholaminergic regions of a mouse model. DA accumulation was associated with a reduced level of GSH, increased production of hydrogen peroxide, as well as enhanced amounts of DA metabolites and cysteinyl-DA adducts [29]. Consistent with these results, higher levels of cysteinyl DA adducts were found in the PD substantia nigra of post-mortem brain samples in comparison to those of the control subjects, although the authors pointed out that their observation could have arisen from the L-DOPA treatment the PD patients took to alleviate their symptoms [30].

The mechanisms of DA-induced cell death have been widely studied for a long time in several cell models [31,32,33,34,35,36,37,38,39]. Both ROS and toxic quinones have been implicated in DA-induced cell death, which can be effectively prevented by NAC, ascorbate, and other antioxidants. Recently, using SH-SY5Y human neuroblastoma cell lines as an experimental model, the effects derived from the DA-related production of ROS were discriminated from those mediated by the presence of DAQs. Interestingly, in the experimental conditions tested, cellular damage due to DA-derived ROS production appeared to be more relevant from a quantitative point of view than the toxic effects induced by DAQs were [40]. On the other hand, DAQs instead of ROS were reported to be primarily involved in causing mitochondrial dysfunction and cell death, with clear evidence of apoptosis in PC12 cells, a rat pheochromocytoma cell line, exposed to DA [34]. It is possible that the nature of the cell lines used in the experiments, as well as the concentrations of DA and the duration of treatments, may determine whether ROS or DAQs would be the major damaging species in a given context. It is worth mentioning that since DAQs are exclusively produced in dopaminergic neurons, they could accumulate over a long period of time and contribute to the preferential degeneration of this neuronal population. In fact, while ROS production is a general process that may occur in every type of cell, DAQ formation is a unique feature of dopaminergic neurons.

## 4. Protein Targets of DAQs

DAQs-related reactivity has been analyzed in depth in the past and in recent years, and numerous protein targets have been identified. It is interesting to note how some of them are directly involved in DA synthesis and cytosolic accumulation, such as TH and DAT [41,42,43], suggesting the presence of a vicious auto-propagating cycle originating from an alteration in DA metabolism that further boosts this impairment. In the first studies, several researchers focused their attention on the reactivity of DAQs toward the proteins associated with familial forms of PD. The rationale behind these investigations was that DAQs-mediated modifications could induce dysfunctional effects such as those derived from the PD-linked pathological mutations on the very same proteins.

Through a proteomic analysis carried out on isolated rat brain mitochondria and differentiated SH-SY5Y cells incubated with radioactive ^14^C-DA, DAQs were found to react with DJ-1, a protein that exerts a protective role against oxidative damage [44]. The structural details of this interaction were then further characterized [45]. In two different cellular models, DAQs have been also demonstrated to react with parkin, an E3 ubiquitin ligase that participates in mitochondrial quality control, promoting its aggregation and inhibiting its ubiquitin ligase activity, therefore suggesting an impairment in proteasomal functionality [46]. Consistent with this result, proteasomal inhibition was observed by treating rabbit reticulocyte lysates or dopaminergic MN9D cells with increasing amounts of DAQs [47,48].

Considering the pivotal role that αSyn plays in the pathogenesis of PD, a number of studies were conducted to investigate the effects of DAQs on the protein, with results that were contradictory. The interaction between αSyn and DAQs was first that was shown to be able to inhibit αSyn fibril formation by stabilizing oligomeric species and promoting the formation of toxic annular protofibrils [49,50]. However, other investigators demonstrated that following interaction with DAQs, αSyn retains an unstructured conformation forming non-amyloidogenic soluble aggregates [51,52,53]. Another controversial aspect was related to the covalent or non-covalent nature of αSyn and DAQs adducts. While several studies identified covalent modifications on αSyn induced by DAQs [50,52,53,54], convincing data also indicated the formation of non-covalent interaction between DAQs and the 125-YEMPS-129 region of αSyn [55,56]. The coexistence of these two situations appears to be the most likely scenario, with the covalent adducts representing only a small fraction of the total number of proteins. As discussed below in another section, considering that αSyn does not possess any cysteine residue, the main target of DAQs’ reactivity, recent studies focused on the interaction between the protein and DOPAL, another DA-derived metabolite, whose interaction with αSyn has been suggested to be more pathologically relevant [57,58].

Additionally, the inactivation of glutathione peroxidase 4 (GPx4) by DAQs was reported earlier with the purified enzyme, as well as in rat brain mitochondrial preparation [59]. GPx4 inhibition is considered to be a major trigger of a special type of ROS-dependent regulated cell death called ferroptosis, which is currently under intense investigation as a possible mechanism of PD neurodegeneration. However, another study reported that DA inhibits erastin-induced ferroptotic death in both cancer and non-cancer cells through multiple mechanisms [60].

Although most studies focused on the interaction of DAQs with specific proteins, a few of them adopted a more general approach to detect a broad range of interactors. Through proteomic studies, Hastings’s group identified both mitochondrial and endoplasmic reticulum (ER) proteins that were affected following exposure to DAQs [44,61,62]. While mitochondrial involvement in DA-related toxicity has been largely investigated and is described in the following section, ER stress is more elusive, but most probably, it is equally important. In fact, a more recent proteomic analysis performed in SH-SY5Y cells using a DA-mimetic containing a biorthogonal alkyne group found several pathways affected by DAQs modifications, which include ER stress, cytoskeletal instability, proteotoxicity, and clathrin function. Through competitive experiments with recombinant protein and in cellular lysates, the investigators then confirmed the functional inhibition by DAQs of the protein disulfide isomerase 3, providing further evidence that dysregulated cytosolic DA levels may induce or exacerbate ER stress [63].

As already mentioned above, the other pathways involved in PD that could be affected by DAQs reactivity are the proteasomal and lysosomal degradative pathways, whose impairment may induce proteostasis alterations that, in the context of PD, are mainly related to αSyn accumulation and aggregation, eventually leading to LB formation. In addition to the previously cited studies related to the proteasome, another work demonstrated that DAQs modification on αSyn inhibited the chaperone-mediated autophagy (CMA) pathway for other substrates [64], replicating the effects observed with other pathological PD-associated αSyn mutants [65]. More specifically, using isolated lysosomes, the investigators observed that DAQ-modified αSyn binds tightly to the lysosomal membrane, but it does not efficiently translocate into the lysosomal lumen, also blocking the uptake of other CMA substrates [64].

More recently, using induced pluripotent stem cells derived from PD-patient fibroblasts and differentiated into dopaminergic neurons, a toxic cascade was observed, in which mitochondrial oxidative stress led to the accumulation of oxidized DA, lysosomal impairment, and αSyn accumulation, linking together DA oxidation with mitochondrial and lysosomal dysfunction [66].

## 5. DAQs’ Reactivity toward Mitochondria

As it is probably due to the fact that oxidative stress and mitochondrial dysfunction are strictly correlated in the context of PD and that DA cytosolic accumulation can result in increased oxidative conditions, the relationship between DA oxidative chemistry and mitochondrial impairment has been largely investigated.

In pioneering work, researchers investigated the effects of DA and DAQs on isolated rat brain mitochondria. While incubation with DA impaired state 3 respiration, an effect that was shown to derive from the MAO-dependent H_2_O_2_ production, DAQs strongly stimulated state 4 respiration, indicating an increase in the rate of proton leakage across the membrane, with respiration that is not used to produce ATP [67]. Mitochondrial swelling due to the opening of the permeability transition pore (mPTP) was also observed in the presence of DAQs, but not with DA alone. Similar effects were subsequently described by other groups [68,69,70]. Consistent with these results, the impairment in the electron transport chain (ETC) activity, as assessed by measuring both complex I and complex IV activities, has been also described in the presence of low concentrations of DA (0.1–0.4 mM) and prolonged exposure periods (up to 2 h) [71,72]. Confirming the central role of DAQs in altering mitochondrial functionality, the effects were strongly enhanced by adding DA in the presence of tyrosinase, which causes its rapid oxidation into DAQs, while they were almost inhibited in the presence of reduced glutathione, a DAQs scavenger. Additionally, a significant amount of DAQs and quinoprotein adducts were observed during the incubation of mitochondria with DA [72]. With the aim of better understanding the mechanism through which DAQs can promote the opening of the mPTP, it has been then demonstrated that their presence leads to the oxidation of NADH in the mitochondrial matrix, suggesting that DAQs toxicity at the level of mitochondria, involves, at least in part, the oxidation of NADH [68]. Importantly, besides promoting the opening of the mPTP, NADH oxidation could also affect the activity of the ETC, especially at the level of complex I. In fact, the electrons derived from NADH oxidation by complex I participate in the generation of the proton gradient used by the ATP synthase to produce ATP.

While most studies used isolated mitochondria to evaluate the effects of DAQs, analyses were also performed in a more physiologically relevant cellular environment [31,32,34]. Overall, they confirmed a decreased level of activity of the mitochondrial respiratory chain complexes, with a higher inhibition observed for complex I and complex IV in comparison to those of complexes II–III, which were associated with a concomitant reduction of the amount of cellular ATP. Moreover, the loss of mitochondrial membrane potential and the opening of the mPTP were also observed, and, consistent with mPTP opening as the underlying cause of depolarization, cyclosporin A was able to protect mitochondria from depolarization. Finally, alterations in mitochondrial morphology were also described confirming the well-accepted idea that mitochondrial morphology and functionality are strictly correlated. More specifically, the elongated tubular morphology observed in the control became less prominent after the treatment with DAQs, with the appearance of fragmented and doughnut-shaped mitochondria.

## 6. DOPAL Toxicity: The Catecholaldehyde Hypothesis

As previously mentioned, the catabolism of cytosolic DA starts in the outer mitochondrial membrane, where the neurotransmitter is transformed in DOPAL by the action of monoamine oxidase, in a reaction that also produces H_2_O_2_. The enzyme aldehyde dehydrogenase is then converted DOPAL into DOPAC, which can rapidly exit the cell. As recently described in exhaustive reviews [57,58,73], DOPAL is a highly reactive molecule, and its cellular accumulation could account for the selective death of dopaminergic neurons, according to the so-called catecholaldheyde hypothesis. In this context, DOPAL exogenous administration in rat substantia nigra resulted in the selective degeneration of dopaminergic neurons and induced locomotor dysfunctions [74,75]. Consistent with the catecholaldheyde hypothesis, exposure to pesticides that are able to inhibit the activity of ALDH has been associated with an increased risk of PD onset [76,77,78]. Moreover, analyses performed on post-mortem PD brains identified the reduced expression of the ALDH1A1 isoform among the molecular determinants involved in the preferential susceptibility of dopaminergic neurons [79,80], further sustaining that the resulting DOPAL accumulation might be among the driving forces for dopaminergic neuron degeneration. Accordingly, a DOPAL buildup has been described in the putamen of PD brains [81,82].

DOPAL reactivity depends on the presence of two functional groups: the catechol moiety and the aldehyde portion. The former one mainly reacts with thiols groups, such as the ones found in cysteines, while the latter one targets primary amines, including the ones of lysine side chains. Because of the presence of the catechol group, DOPAL reactivity overlaps with that of DA. Accordingly, DOPAL has been demonstrated to form adducts with many proteins, including TH, DOPA-decarboxylase, VMAT2, glucocerebrosidase, and ubiquitin [83]. Moreover, in energetically compromised mitochondria, DOPAL has been shown to induce the opening of the mPTP in a much stronger way than DA does [84]. In fact, the presence of the aldehyde group confers to DOPAL an additional reactivity in comparison to DA, which may be extremely relevant for PD pathology, especially when αSyn is taken into consideration. As aforementioned, αSyn has no cysteine residues, making it an unlikely target for DAQs. In contrast, the protein is rich in lysines, which account for more than 10% of its primary sequence, a percentage higher than the average value (around 5%) of the lysine fraction in synaptic proteins [85]. Moreover, considering that αSyn is an intrinsically unfolded protein [86], all the lysine residues are easily accessible for any potential chemical modification. It is also worth mentioning that αSyn represents the 0.5%–1% of the total soluble proteins of the brain, reaching a concentration up to 40 μM in pre-synaptic terminals of neurons [85], where DOPAL is produced following DA catabolism. All these aspects make the interaction between DOPAL and αSyn extremely relevant from a pathological standpoint.

In 2008, Burke and co-workers demonstrated for the first time that DOPAL, at physiologically relevant concentrations, actually promotes αSyn aggregation, both in vitro and in SH-SY5Y human neuroblastoma cells [87]. Moreover, the injection of DOPAL into the rat substantia nigra, in addition to inducing the degeneration of dopaminergic neurons, also promoted the accumulation of high-molecular-weight αSyn oligomers, suggesting a direct relationship between the two phenotypes [87]. Divalent metal cations, especially Cu(II), were then demonstrated to enhance the DOPAL-induced oligomerization of αSyn [88], while the treatment with anti-oxidants, such as reduced glutathione, ascorbic acid, or N-acetylcysteine resulted in an attenuated oligomerization [83,89,90]. Consistent with the aforementioned considerations on DOPAL reactivity, the lysine residues of αSyn have been demonstrated to represent its principal target, and the most reactive lysines have been characterized, although some discrepancies are present between the results obtained in vitro or in a cellular environment [83,90,91].

DOPAL-induced αSyn oligomerization may promote cell toxicity in different ways. First, oligomers have been described to affect mitochondrial functionality by inhibiting their oxygen consumption and decreasing the mitochondrial membrane potential, with effects that appear to be dependent on complex I impairment [92]. Alternatively, oligomers have been proven to form pores on vesicles membrane and to induce a redistribution of synaptic vesicle pools in primary neuronal cultures [91]. More specifically, a strong decrease in the number of vesicles per synapse was observed in neurons treated with DOPAL, with a reduction of the fraction of synaptic vesicles in the ready-releasable pool and an increase in the vesicles size of the resting pool, suggesting that αSyn-DOPAL oligomers may preferentially damage the vesicles ready to be released at the synapse, possibly because oligomer formation occurs in the subcellular region where αSyn is known to participate in synaptic vesicle recycling [91]. More recently, proteostasis impairment has been indicated as a major contributing factor to DOPAL-triggered αSyn neurotoxicity [93]. In particular, DOPAL-induced αSyn accumulation among the neuronal compartments was observed, as well as impaired αSyn clearance in primary neuronal cultures. Investigators also assessed the differential impact of the αSyn-DOPAL interplay in diverse neuronal districts, revealing altered synaptic integrity, overwhelmed degradative pathways in neuronal projections, and reduced axonal arborization [93].

## 7. DA and Neuroinflammation: The Role of Neuromelanin

As underlined in the introduction, a correlation exists between oxidative stress, mitochondrial dysfunction, and neuroinflammation, with αSyn aggregation being able to fuel every single process. In this scenario, DA could contribute to chronic neuroinflammation by promoting both mitochondrial dysfunction and the accumulation and release of αSyn in the extracellular environment. Another way through which DA could mediate the preferential degeneration of dopaminergic neurons observed in PD is represented by NM.

As aforementioned, NM is a dark, insoluble pigment that accumulates with aging inside autolysosomal organelles [94]. It is synthesized from oxidized DA, which reacts with protein fibrils, forming seeds that promote pigment growth. Among the proteins that are able to interact with DAQ, αSyn fibrils were identified in NM-containing organelles of the substantia nigra [94]. Besides DA, proteins, and lipids, NM also contains metal components, the most abundant of which are iron, zinc, and copper [24]. The relationship between metals and NM is quite intriguing. On one hand, iron, and to a lesser extent, copper can promote NM formation through the oxidation of DA [24,95,96]. On the other hand, NM is able to trap metals inside the pigment, protecting cells from the potential toxicity that could take place through metal-related redox reactions [24,25]. These aspects suggest that, under physiological conditions, NM exerts a neuroprotective role by sequestering reactive DAQs and metal ions, as well as toxic αSyn aggregates. Nevertheless, under the pathological events that characterize PD, the situation becomes totally different, and NM might fuel neurodegeneration. In fact, the NM released into the extracellular space from dying dopaminergic neurons may promote neuroinflammation through microglia activation [24,25]. Accordingly, in the substantia nigra of PD patients, the presence of numerous extracellular NM deposits has been described in close proximity to activated microglia as the result of severe degeneration of dopaminergic neurons [97,98]. NM-induced microglia activation, with the release of reactive molecules and proinflammatory cytokines, has been confirmed in cellular cultures [99,100]. All these data converge on the key role of NM in glia activation and the disease’s progression.

## 8. Conclusions

The studies discussed in this review have strongly indicated the importance of DA oxidation products in the pathogenesis of PD. However, while many of the studies dealing with DA toxicity on cells or isolated mitochondria or other organelles have utilized sub-millimolar concentrations of DA, usually from 0.1 to 0.5 mM, the concentration of free cytosolic DA in nigral dopaminergic neurons, which is prone to enzymatic or non-enzymatic oxidation, is not known with certainty in physiological or even pathological conditions such as PD. Thus, the extrapolation of the results of studies carried out with high concentrations of DA relating to the pathogenesis of clinical PD might be erroneous. For instance, it could be possible that low concentrations of DA-derived ROS and DAQs in nigral neurons may cause altered redox signaling instead of direct damage to neuronal components. In turn, the altered functions of ROS-sensitive transcription factors and kinases may affect the viability and functions of neurons or may trigger an inflammatory microglial response. This aspect has not been thoroughly explored in experimental PD research and deserves more attention. Another aspect that should be considered is that in contrast to the substantia nigra, dopaminergic neurons in other brain regions, such as the ventral tegmental area, are less or not affected during the disease’s progression. This observation suggests that in dopaminergic neurons of the substantia nigra, other peculiar risk factors might be present, which could amplify the otherwise tolerable DA-dependent toxicity. In this context, many mechanisms have been proposed, including the presence of highly branched, poorly myelinated long axons, which are associated with mitochondrial oxidative stress, and large fluctuations in cytosolic calcium levels [101]. Another interesting process that has recently emerged in the literature is ferroptosis, an iron-dependent cell death pathway that involves the accumulation of lipid peroxidation accompanied by the concomitant depletion of intracellular GSH and that has been suggested as a contributing mechanism of neuronal death in PD [102,103,104]. This process is regulated in a complex way through the involvement of multiple enzymes, iron-binding proteins, and several transport systems. Thus, the potential contribution of DA oxidation products in triggering ferroptosis in PD experimental models could be extremely relevant and should be critically examined. As aforementioned, PD is still an incurable disorder and besides L-DOPA administration, the currently available therapeutic approaches focus on the stimulation of dopaminergic signaling, such as DA agonists and inhibitors of the DA degradative pathways. While these drugs are often used as adjuvants to L-DOPA treatment to alleviate long-term motor complications, some of them, such as the MAO inhibitors selegiline and rasagiline, have also been used as a monotherapy in the early stages of the disease [105]. In addition to preventing DOPAL accumulation, these drugs have been described to promote the expression of several neurotrophic factors and to inhibit both apoptosis and αSyn-aggregation, with rasagiline possibly having a therapeutic advantage over selegiline due to a different catabolic process [105]. Unfortunately, none of the existing treatments are able to stop the progression of the disease, and the reason most probably relies on the fact that DA-associated toxicity strictly depends on its cellular metabolism, which cannot be modified without affecting physiologically relevant processes based on this neurotransmitter. For this reason, a multi-target strategy could be more effective in hampering PD progression, and more knowledge of the cellular toxicity associated with DA and of the risk factors able to amplify such a DA-dependent toxicity could reveal novel promising therapeutic targets.

## Figures and Tables

**Figure 1 antioxidants-12-00955-f001:**
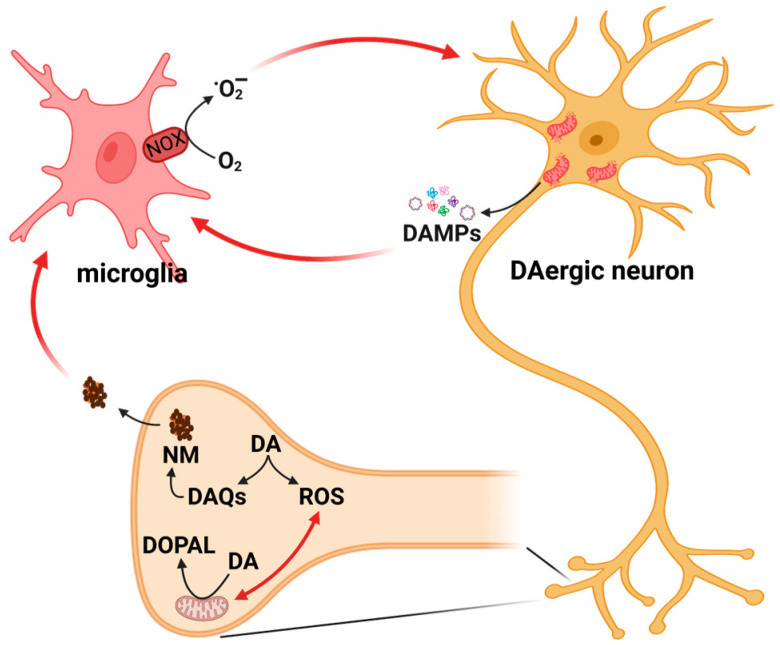
The vicious cycle in Parkinson’s disease, involving alterations in redox homeostasis, dysfunctional mitochondria, and neuroinflammation. Mitochondria are the main site of ROS production, and, at the same time, ROS may affect mitochondrial functionality. Mitochondria can stimulate neuroinflammatory responses through the release of damage-associated molecular patterns (DAMPs). Chronic activation of microglia stimulates the production of ROS through the activation of NADPH oxidase (NOX). In dopaminergic neurons, cytosolic dopamine (DA) can auto-oxidase leading to the generation of ROS and reactive quinones (DAQs), which can either react with cellular nucleophiles or aggregate into neuromelanin (NM). The release of NM from dying neurons can fuel neuroinflammatory processes by activating microglia. Alternatively, cytosolic DA can be metabolized at the mitochondrial level into the very reactive molecule dihydroxyphenylacetaldehyde (DOPAL). The role of the oxidation product of DA in this vicious cycle will be discussed in the following sections. (Created with BioRender.com, accessed on 13 April 2023.)

**Figure 2 antioxidants-12-00955-f002:**
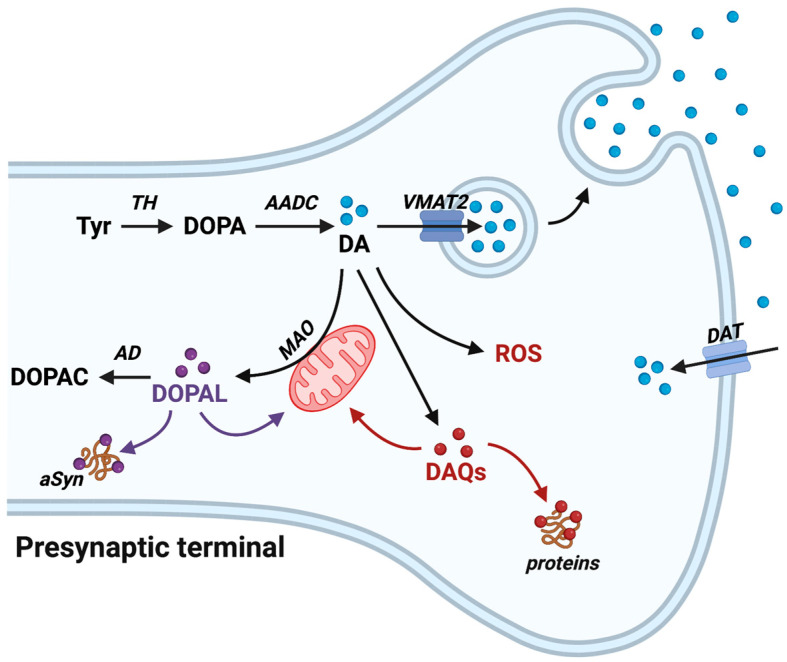
The pathological implication of dopamine (DA) metabolism. DA is synthesized in the cytoplasm by the action of tyrosine hydroxylase (TH) and aromatic amino acid decarboxylase (AADC), and it is rapidly sequestered into synaptic vesicles by the vesicular monoamine transporter 2 (VMAT2), where it is stabilized by the low pH. Following its release into the synaptic cleft, DA reuptake rapidly ensues via the DA transporter (DAT). If the amount of cytosolic DA exceeds the physiological concentration, it can be metabolized via monoamine oxidase (MAO) into 3,4-dihydroxyphenilacetldehyde (DOPAL), a highly toxic molecule that, in turn, is converted to the non-toxic metabolite 3,4-dihydroxyphenylacetic acid (DOPAC) by the enzyme aldehyde dehydrogenase. At cytosolic pH, DA can auto-oxidize leading to the formation of ROS and reactive quinones (DAQs). Both DAQs and DOPAL can react with cytosolic nucleophiles, such as proteins, affecting their functionality and inducing mitochondrial impairment. The interaction between DOPAL and α-synuclein (αSyn) seems to contribute to αSyn-neurotoxicity in Parkinson’s disease. (Created with BioRender.com, accessed on 1 March 2023.)

## Data Availability

Not applicable.

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
