# Peer review of "Oxidative Stress and Neuroinflammation in Parkinson’s Disease: The Role of Dopamine Oxidation Products"

_antioxidants, 2023, doi:10.3390/antiox12040955_

Round 1

Reviewer 1 Report

antioxidants-2337950: "Oxidative stress and neuroinflammation in Parkinson’s disease: the role of dopamine oxidation products"

In this comprehensive review, the authors analyze various subcellular mechanisms involved in Parkinson’s disease development. All materials are described successively and intelligibly; they evidently demonstrate the crucial role of oxidative stress and neuroinflammation in this neurodegenerative pathology. The manuscript could be published even in the current form; however, the text might be improved by the expanding of the “DA and neuroinflammation: the role of neuromelanin” section at the expence of materials associated with senescent cells.

Comment: In Figure 1, abbreviations of “ROS”, “DAMPs”, and “NADPH oxidase” would be desirable to place. 

Author Response

In this comprehensive review, the authors analyze various subcellular mechanisms involved in Parkinson’s disease development. All materials are described successively and intelligibly; they evidently demonstrate the crucial role of oxidative stress and neuroinflammation in this neurodegenerative pathology. The manuscript could be published even in the current form; however, the text might be improved by the expanding of the “DA and neuroinflammation: the role of neuromelanin” section at the expence of materials associated with senescent cells.

We thank the referee for the positive comments. We agree that the involvement of cellular senescence in neurodegenerative disorders, including PD, is becoming more and more evident. While this topic certainly deserves attention as certified also by recent reviews that have been published (PMID: 34366147; 35661020; 35721008), we think that it is out of the aim of our review.

Comment: In Figure 1, abbreviations of “ROS”, “DAMPs”, and “NADPH oxidase” would be desirable to place. 

A new Figure 1 has been prepared, which includes the referee’s suggestions.

Reviewer 2 Report

The manuscript Antioxidants-2337950, entitled: “Oxidative stress and neuroinflammation in Parkinson’s disease: the role of dopamine oxidation products” by Chakrabarti and Bisaglia, is a well documented review of the neurochemical studies involving dopamine in Parkinson disease, and in my opinion is deserving publication.            

MINOR OBSERVATIONS

1) The authors are briefly pointing out in the conclusion that many data in the literature proving dopamine-related neurodegeneration have been obtained using very high concentrations of dopamine, that may not be occurring in Parkinson’s disease patients. The idea that dopamine metabolism may be the cause of Parkinson’s disease is quite old and established, but this does not mean that it must be completely true. One important aspect is the occurrence of damage in the substancia nigra in the absence of damage of the ventral-tegmental area, which dopaminergic neurons are responsible for a proper functioning of both the mesolimbic and the mesocortical pathways. Albeit a certain number of Parkinson’s disease patient do develop a cognitive disfunction that may imply an impairment of the ventral tegmental area, others do not. Thus, the selective fragility of the dopaminergic neurons of the substancia nigra of these patients may be based on a peculiar mechanism that may amplify the otherwise tolerable dopaminergic insult. Individual variability in gene expression and RNA editing may be conditioning the response to oxidative stress (Garcia et al. 2021, doi:10.2174/1567205018666211117101216).

2) A mention to the observed delay in the progression of experimental Parkinson’s disease with selegiline and rasagiline should be mentioned. These drugs do not seem to work similarly in humans, and this may further support the idea that idiopathic Parkinson’s disease is a progressive neurodegenerative process occurring in a specific neuronal population via mechanisms that cannot be fully mimicked in animals that do not develop idiopathic Parkinson’s disease.

Author Response

The manuscript Antioxidants-2337950, entitled: “Oxidative stress and neuroinflammation in Parkinson’s disease: the role of dopamine oxidation products” by Chakrabarti and Bisaglia, is a well documented review of the neurochemical studies involving dopamine in Parkinson disease, and in my opinion is deserving publication. 

We thank the referee for the positive comments.

MINOR

1) The authors are briefly pointing out in the conclusion that many data in the literature proving dopamine-related neurodegeneration have been obtained using very high concentrations of dopamine, that may not be occurring in Parkinson’s disease patients. The idea that dopamine metabolism may be the cause of Parkinson’s disease is quite old and established, but this does not mean that it must be completely true. One important aspect is the occurrence of damage in the substancia nigra in the absence of damage of the ventral-tegmental area, which dopaminergic neurons are responsible for a proper functioning of both the mesolimbic and the mesocortical pathways. Albeit a certain number of Parkinson’s disease patient do develop a cognitive disfunction that may imply an impairment of the ventral tegmental area, others do not. Thus, the selective fragility of the dopaminergic neurons of the substancia nigra of these patients may be based on a peculiar mechanism that may amplify the otherwise tolerable dopaminergic insult. Individual variability in gene expression and RNA editing may be conditioning the response to oxidative stress (Garcia et al. 2021, doi:10.2174/1567205018666211117101216).

We thank the referee for her/his comments. According to the referee’s suggestions, in the revised version of the manuscript we have added a new paragraph in the conclusions (lines #422-430).

2) A mention to the observed delay in the progression of experimental Parkinson’s disease with selegiline and rasagiline should be mentioned. These drugs do not seem to work similarly in humans, and this may further support the idea that idiopathic Parkinson’s disease is a progressive neurodegenerative process occurring in a specific neuronal population via mechanisms that cannot be fully mimicked in animals that do not develop idiopathic Parkinson’s disease.

According to the reviewer’s suggestion, in the revised version of the manuscript, we have added a new paragraph that mentions the use of rasagiline and selegiline in PD treatment (lines #442-448).

Reviewer 3 Report

This well-written review article summarizes the latest studies about the relationships between dopamine oxidative products, oxidative stress, and neuroinflammation. Here are my several suggestions before being accepted for publication:

1.      In Figure 1, the authors may consider drawing a dopamine neuron and indicate the mitochondria inside the DA neuron. It would be more reasonable to see the cell-cell interaction, not the cell-organelle interaction. Please also label microglia, DAMPs in the figure.

1.      I would suggest the authors include a paragraph discussing the current treatment of PD using levodopa. Levodopa is the dopamine precursor, but studies found that levodopa has toxicity to neurons, which may be due to oxidative stress. This would be worth discussing in this article.

3.      The authors may consider writing a future direction section in their conclusion.

Author Response

This well-written review article summarizes the latest studies about the relationships between dopamine oxidative products, oxidative stress, and neuroinflammation. Here are my several suggestions before being accepted for publication:

1.   In Figure 1, the authors may consider drawing a dopamine neuron and indicate the mitochondria inside the DA neuron. It would be more reasonable to see the cell-cell interaction, not the cell-organelle interaction. Please also label microglia, DAMPs in the figure.

A new Figure 1 has been prepared, which includes the referee’s suggestions.

2.   I would suggest the authors include a paragraph discussing the current treatment of PD using levodopa. Levodopa is the dopamine precursor, but studies found that levodopa has toxicity to neurons, which may be due to oxidative stress. This would be worth discussing in this article.

According to the reviewer’s suggestion, in the revised version of the manuscript, we have added a new paragraph that mentions the use of levodopa in PD treatment and the long-term associated side effects. (lines #39-47).

3.   The authors may consider writing a future direction section in their conclusion.

According to the reviewer’s suggestion, at the end of the conclusions, a new paragraph has been added (lines #439-455).